# Antioxidant and Polyphenol-Rich Ethanolic Extract of *Rubia tinctorum* L. Prevents Urolithiasis in an Ethylene Glycol Experimental Model in Rats

**DOI:** 10.3390/molecules26041005

**Published:** 2021-02-14

**Authors:** Fatima Zahra Marhoume, Rachida Aboufatima, Younes Zaid, Youness Limami, Raphaël E. Duval, Jawad Laadraoui, Anass Belbachir, Abderrahmane Chait, Abdallah Bagri

**Affiliations:** 1Laboratory of Biochemistry and Neuroscience, Integrative and Computational Neuroscience Team, Faculty of Sciences and Technology, Hassan First University, Settat 26002, Morocco; marhoume.fatimazahra@gmail.com; 2Laboratory of Neurobiology, Pharmacology and Behavior, Faculty of Sciences Semlalia, Cadi Ayad University, Marrakech 40000, Morocco; jawad.laadraoui@edu.uca.ac.ma (J.L.); chait@uca.ma (A.C.); 3Laboratory of Biological Engineering, Faculty of Sciences and Technology, Sultan Moulay Slimane University, Beni Mellal 23000, Morocco; raboufatima@gmail.com; 4Research Center of Abulcasis University of Health Sciences, Rabat 10000, Morocco; younes_zaid@yahoo.ca (Y.Z.); youness.limami@gmail.com (Y.L.); 5Botany Laboratory, Biology Department, Faculty of Sciences, Mohammed V University in Rabat, Rabat 10000, Morocco; 6Immunology and Biodiversity Laboratory, Department of Biology, Faculty of Sciences, Hassan II University, Casablanca 20000, Morocco; 7Université de Lorraine, CNRS, L2CM, F-54000 Nancy, France; 8Morpho-Science Research Laboratory, Faculty of Medicine and Pharmacy, Cadi Ayad University, Marrakech 40000, Morocco; belbachir@gmail.com; 9Regenerative Medicine Center University Hospital Center of Mohammed VI Marrakech, Marrakech 40000, Morocco

**Keywords:** *Rubia tinctorum* L., antioxidants, polyphenols, ethylene glycol, urolithiasis, histophatology

## Abstract

Treatment of kidney stones is based on symptomatic medications which are associated with side effects such as gastrointestinal symptoms (e.g., nausea, vomiting) and hepatotoxicity. The search for effective plant extracts without the above side effects has demonstrated the involvement of antioxidants in the treatment of kidney stones. A local survey in Morocco has previously revealed the frequent use of *Rubia tinctorum* L. (RT) for the treatment of kidney stones. In this study, we first explored whether RT ethanolic (E-RT) and ethyl acetate (EA-RT) extracts of *Rubia tinctorum* L. could prevent the occurrence of urolithiasis in an experimental 0.75% ethylene glycol (EG) and 2% ammonium chloride (AC)-induced rat model. Secondly, we determined the potential antioxidant potency as well as the polyphenol composition of these extracts. An EG/AC regimen for 10 days induced the formation of bipyramid-shaped calcium oxalate crystals in the urine. Concomitantly, serum and urinary creatinine, urea, uric acid, phosphorus, calcium, sodium, potassium, and chloride were altered. The co-administration of both RT extracts prevented alterations in all these parameters. In the EG/AC-induced rat model, the antioxidants- and polyphenols-rich E-RT and EA-RT extracts significantly reduced the presence of calcium oxalate in the urine, and prevented serum and urinary biochemical alterations together with kidney tissue damage associated with urolithiasis. Moreover, we demonstrated that the beneficial preventive effects of E-RT co-administration were more pronounced than those obtained with EA-RT. The superiority of E-RT was associated with its more potent antioxidant effect, due to its high content in polyphenols.

## 1. Introduction

Urolithiasis, or kidney stones, is a major health concern with increasing prevalence rates worldwide [1]. It results from free or attached mineral crystallizations in the renal calyces [2]. The crystalline and organic components are formed when the urine becomes supersaturated with minerals, then they grow, and aggregate before being retained [3]. About 80% of calculi are composed of calcium phosphate (CaP) mixed with calcium oxalate (CaOx) [4]. Multiple factors can cause urolithiasis, including a sedentary lifestyle, unhealthy diet, irregular food habits and obesity [5]. the treatment of kidney stones involves the administration of symptomatic drugs (diuretics, anti-inflammatory drugs), percutaneous, nephrolithotomy and lithotripsy [6]. However, these treatments are frequently associated with complications such as hemorrhage, hypertension, and tubular necrosis followed by subsequent fibrosis [7]. In addition, they are very expensive and to date, there is no promising drug for the treatment and prevention of recurrence [8].

Herbal remedies could be an alternative to anti-urolithiasic drugs due to their many active compounds that can act synergistically and have—most the time—minimal side effects [9]. It was suggested that plants with anti-urolithiasic properties induce their effect via antioxidant capacities that mitigate the toxicity caused by free radicals involved in the initiation and development of urolithiasis [9]. Many experimental studies support this hypothesis. Indeed, phenols and flavonoids have been shown to be effective in attenuating the process of calculus formation, both in animal models and in humans [10]. Polyphenols from grape seeds prevented renal papilla from calcium monohydrate oxalate, calculi formation and lesions induced by oxidant cytotoxic substances [11]. Furthermore, polyphenols-rich extract from *Quercus gilva* Blume showed an anti-urolithiasic effect associated with its antioxidant and anti-inflammatory properties [12].

*Rubia tinctorum* L. (madder root) (RT) is a plant belonging to the Rubiaceae family whose root is used as a folk medicine to cure various ailments, including kidney stones and bladder diseases in several countries in Asia, Russia and Europe [9,13]. The therapeutic properties of RT such as anti-inflammatory, antioxidant, hepatoprotective and antibacterial activities were confirmed in vivo and in vitro by experimental data [14]. A local survey in Morocco has previously revealed the frequent use of *Rubia tinctorum* L. (RT) for the treatment of kidney stones [15]. However, to our knowledge, no experimental study reported an anti-urolithiasic effect of RT. Therefore, the present study was conducted on an ethylene glycol (EG)/ammonium chloride (AC) experimental model of urolithiasis in rats. The aim of this study was to evaluate the potential protective effect of both ethanolic and ethyl acetate extracts of *Rubia tinctorum* L. (E-RT and EA-RT, respectively) in this model. Then, the objective was to assess the RT extract’s antioxidant activity and to identify the polyphenols contained in E-RT and EA-RT extracts which could be linked to these effects.

## 2. Results

### 2.1. Pathophysiology of EG/AC-Induced Urolithiasis

The treatment of rats for 10 days with EG/AC resulted in urolithiasis formation and in alterations in urinary, serum and kidney tissue parameters. Analysis of the changes induced showed that our experimental design reproduced the pathophysiology of a form of urolithiasis in the rat. As shown in Table 1, the levels of serum urea, creatinine, uric acid, calcium and phosphorus of the lithiasic group (G2) were significantly increased (5.91 ± 0.17, 39.51 ± 4.86, 44.86 ± 3.00, 119 ± 1.59 and 107.16 ± 1.74, respectively) compared to the vehicle control group (G1) (0.26 ± 0.02, 5.49 ± 0.21, 10.71 ± 0.29, 103.77 ± 1.01 and 83.34 ± 0.33, respectively). The levels of urinary urea, creatinine, uric acid, calcium and phosphorus concentrations were also significantly increased in the urolithiasic group (317.90 ± 0.95, 116.28 ± 4.02, 230.5 ± 10.79, 217.08 ± 3.73 and 2169.41 ± 1.47, respectively) when compared to G1 (33.96 ± 0.95, 27.75 ± 1.66, 58.33 ± 3.48, 74.36 ± 1.36 and 12.83 ± 0.60, respectively) (Table 2). Similar changes were also observed for electrolyte concentrations in urine and serum (e.g., Na^+^, Cl^−^, K^+^) (Table 1 and Table 2). Indeed, the urolithiasic group’s urine contained significant higher concentrations of potassium, chloride and sodium (196.03 ± 0.81, 132.31 ± 0.75 and 18.33 ± 0.49, respectively) than G1 (6.13 ± 0.16, 36.45 ± 0.59 and 8.33 ± 0.21 respectively). Moreover, marked histological alterations (Table 3), including interstitial mononuclear cell infiltration and damage in glomeruli, were also observed in the urolithiasic group but not in the vehicle control group.

### 2.2. Effect of RT Extracts on the Pathophysiology of EG/AC-Induced Urolithiasis

#### 2.2.1. Body Weight

The mean body weights (MBW) of the six experimental groups (i.e., vehicle control (G1), lithiasic control (G2), E-RT 1 g/kg (G3), E-RT 2 g/kg (G4), EA-RT 1 g/kg (G5), EA-RT 2 g/kg (G6)) were similar at the beginning of the experiment (day 1). After 10 days of treatment, the MBW of five groups (G2, G3, G4, G5 and G6) were decreased when compared to (G1) (Figure 1). Figure 1 also shows that the group G2 (i.e., −41.08 g) had a greater decrease in MBW than the RT-treated groups (G3, G4, G5 and G6) (i.e., −13.52 g; −5.68 g; −28.53 g; −22.96 g, respectively) and that the MBW of the group treated with E-RT (1 and 2 g/kg) did not significantly differ from the untreated control G1.

#### 2.2.2. Urine Microscopic Analysis

Microscopic analysis of the urine showed that the vehicle control group did not contain CaOx crystals (Appendix A). However, EG/AC treatment resulted in bipyramidic CaOx crystal formation (Appendix A). Concomitant treatment with both RT extracts, regardless of dose, decreased the number and size of the crystals in a dose-dependent manner (Appendix A).

#### 2.2.3. Serum and Urinary Analysis

As mentioned above, serum urea, creatinine, uric acid, calcium, phosphorus, chloride, sodium, and potassium levels were significantly higher in the lithiasic control group (G2) than in the vehicle control group (G1). These levels were significantly reduced by the preventive treatment of the two doses of E-RT: 1 g/kg and 2 g/kg and the higher dose of EA-RT (2 g/kg). The high dose of E-RT was found to be the most effective (Table 1).

In the same way, urinary concentrations of urea, creatinine, uric acid, calcium, phosphorus, chloride, sodium, potassium, and proteins were significantly higher in the lithiasic control group than in the vehicle control group. The treatment with E-RT and EA-RT significantly prevented an EG/AC-induced increase in the level of these concentrations. The high dose of E-RT (G4) was found to be the most effective fraction (Table 2).

#### 2.2.4. Histopathological Analysis

Kidney sections from the vehicle control group (G1) showed normal cell structures in the glomeruli as well as in the renal tubules (T) and blood vessels (Figure 2(G1) and Table 3) while those obtained from the lithiasic control group (G2) showed severe and extensive tissue lesions. The alterations consisted of interstitial infiltration of the mononucleated cells, tubular dilatation, and lesions of the glomeruli (Figure 2(G2) and Table 3). The two doses (1 and 2 g/kg) of E-RT prevented these tissue alterations. Thus, only minor tissue damages in the blood vessels and in the tubular epithelial cells, but not in the glomeruli, were observed in the kidney sections from the E-RT groups (Figure 2 panel G3 and G4). EA-RT (1 and 2 g/kg) was also effective in preventing tissue damage that could be induced by the EG/AC treatment. Kidney histological sections from rats treated with EA-RT showed normal cell structures in the renal tubules, blood vessels, and glomeruli (Figure 2 panel G5 and G6) Furthermore, both RT extracts prevented the mononucleated cells’ infiltration and hemorrhage but E-RT was more efficient than EA-RT (Table 3).

### 2.3. Antioxidant Properties of E-RT and EA-RT and Their Polyphenols Composition

#### 2.3.1. Antioxidant Activity

Antioxidant activity of E-RT and EA-RT was assessed both in vitro (Table 4 and Table 5) and in vivo (Figure 3).

In vitro, antioxidant activity was evaluated by DPPH (Table 4), the reducing power and β-carotene linoleic acid methods (Table 5). The antioxidant activity was determined by the value of the concentration (μg/mL) producing a 50% inhibition (IC_50_) of free radicals. RT extracts showed higher IC_50_ than gallic acid (64.5 ± 0.70 µg/mL) indicating that these extracts possess antioxidant activity that is lower than that of gallic acid (GA). IC_50_ of E-RT was 156.44 ± 35.76 µg/mL whereas IC_50_ of EA-RT was 206.23 ± 90.68 µg/mL, indicating that E-RT antioxidant potential was higher than that of EA-RT (Table 4).

As far as the reducing power and β-carotene linoleic acid methods are concerned, our data document that E-RT exhibited interesting IC_50_ antioxidant activity, respectively 2.44 ± 0.02 and 75.61 ± 3.33. IC_50_ of EA-RT was 4.66 ± 0.04 in reducing power assay, and 101.64 ± 5.41 for β-carotene assay. Overall, the antioxidant activity of the E-RT was higher than EA-RT. However, even if E-RT has an interesting antioxidant activity, it is less important than reference antioxidant agents such butylated hydroxytoluene (BHT) and quercetin.

In vivo, the antioxidant activity was evaluated in kidney homogenates using lipid peroxidation determination; malondialdehyde (MDA) concentration and catalase (CAT) activity (Figure 3).

EG/AC treatment increased significantly the level of MDA (4.60 ± 0.10 µmol/g of tissue) (Figure 3A) and decreased significantly the activity of catalase enzyme (5.35 ± 0.25 U/g of tissue) (Figure 3B) in kidneys of the untreated G2 compared to the vehicle control (G1, 1.75 ± 0.15 µmol/g and 18.85 ± 0.65 U/g, respectively). Treatments with *Rubia tinctorum* L. extracts (E-RT and EA-RT) reduced lipid peroxidation and protected against the oxidative stress induced by EG/AC treatment in G4 and G6 (2.25 ± 0.15 and 2.85 ± 0.05 µmol/g of tissue, respectively). CAT activity was significantly increased in the same groups (G4 and G6) (15.9 ± 0.20 and 17.2 ± 0.30 U/g of tissue, respectively).

#### 2.3.2. Total Phenols Contents

The total phenol contents, determined in a sample volume of 20 µL by the Folin–Ciocalteu method, in the E-RT and EA-RT were 18.37 ± 0.58 and 0.61 ± 0.01 mg of GAE (gallic acid equivalent), respectively. The higher phenolic compound content found in E-RT is statistically different from that found in EA-RT.

#### 2.3.3. Analysis of E-RT and EA-RT Polyphenols Contents

The polyphenol content of each extract, E-RT and EA-RT, respectively, were analyzed using HPLC (Appendix A). On the basis of standard compounds’ retention times (RT, in min), the polyphenols identified in E-RT and EA-RT were syringic acid, rutin, ferulic acid, vanillin, rosmarinic acid, cinnamic acid, catechin, and quercetin. The respective amounts of each compound ranged from 12.14 to 26.74 mg/GAE/100 g DM (dry matter) (Table 6), according to the following order: vanillin > rosmarinic acid > quercetin. In EA-RT, quercetin was the most represented.

## 3. Discussion

The results of the present study show that in response to 0.75% ethylene glycol (EG)/2% ammonium chloride (AC) oral administration over a 10-day period, young male rats developed kidney stones (or urolithiasis) mainly composed of calcium oxalate (CaOx). As shown in the Figure 1, treated rats (urolithiasis control group or G2) lost body weight because they drank less water and almost stopped eating. The EG/AC model used in the present this study is similar to the previously described experimental models of urolithiasis by Ravindra and colleagues [16]. The pathophysiological mechanisms responsible for the alterations elicited in this model could be related to an increase in the urinary oxalate (Ox) concentration. Indeed, EG is easily absorbed along the intestine and metabolized in the liver to Ox, leading to hyperoxaluria. Ox precipitates in the urine as CaOx because of its low solubility. High levels of Ox and CaOx crystals, particularly in the epithelial cells of the nephron, induce heterogeneous nucleation followed by crystal aggregation [17]. AC potentiates the action of EG and accelerates the phenomenon of urolithiasis [18].

Microscopic examination clearly showed that the treatment inducing urolithiasis led to the appearance of characteristic crystals of CaOx (i.e., with a bipyramid form) in the urine, while the urine of the untreated control group (G1) was free of these crystals. This result is similar to previously published data [16] and may be associated with a decreased urinary output, an elevated pH, hyperoxaluria, and hypercalciuria [19]. The biochemical analysis of the urine confirmed these results as lithiasic rats presented an increase in the excretion of phosphorus and calcium. A high concentration of phosphorus in the renal tubules could potentiate the Ox-induced lithiasis [20], whereas calcium would act as an important factor in the nucleation and precipitation of Ox in the form of CaOx [17] and in the resulting crystal growth [20]. An increase in urinary protein excretion has also been recorded indicating proximal tubular dysfunction [21]. Protein excretion could be related to severe lesions of the glomeruli and to tubular dilatation [22]. Another factor contributing to protein urea could be an interstitial inflammation attested by mononucleated cells’ infiltration (Table 3).

Serum levels of urea, creatinine, and uric acid were significantly increased in the urolithiasic group (G2) compared to the untreated control group (G1), indicating renal damage (Table 1). These results are consistent with those of a previous study and indicate that the accumulation of nitrogenous substances in the serum may be a consequence of a decreased glomerular filtration rate (GFR) due to lithiasic obstruction [23]. Uric acid binds to CaOx and modulates its crystallization and solubility and also reduces the inhibitory activity of glycosaminoglycans [20]. Na^+^, K^+^, and Cl^−^ plasma concentrations were significantly increased in the lithiasic group (G2). Electrolyte imbalance disturbs the metabolism of the renal cells leading to the development of cell structure alterations [24].

The EG/AC treatment caused tubular dilation, glomeruli lesions, and mononucleocyte infiltration. These renal damages observed in the lithiasic group (G2) could be attributed to a peroxidative action on the renal epithelium resulting from the elevated rate of urinary Ox and its deposition in the tubules and glomeruli. Indeed, CaOx deposition was shown to induce oxidative stress, which could be responsible for papillary tissue lesions [2].

In the present study, we investigated the antilithiasic activities of ethanol and ethyl acetate extract of *Rubia tinctorum* L. (RT) on EG/AC-induced renal lithiasis in rats. RT is one of the several medicinal plants that are widely used in traditional medicine systems to cure various ailments. The plant has been extensively studied for its biological activities and therapeutic potentials such as anti-inflammatory, anti-aggregant, antioxidant and antibacterial properties [14,25,26]. In traditional medicine, RT dried roots are used for treatment of cardiovascular diseases including high blood pressure [15,27] liver pain, anemia and diarrhea [28,29]. Moreover, several studies have described the nephro-protective effects of plants [9]. However, to our knowledge, no previous studies have demonstrated an anti-urolithiasic activity of RT.

A previous toxicological experiment, conducted in our laboratory, demonstrated that up to 5 g/kg of the RT extracts, administrated orally, triggered no major side effects [26]. Therefore, the selected doses of 1 g/kg and 2 g/kg were free of any toxicological effect.

Treatment with E-RT and EA-RT prevented the alterations induced by EG/AC to values close to the untreated control group (G1). This preventive effect concerned the formation of crystals in the urine and the biochemical parameters of the serum and urine. RT extracts also prevented the body weight loss induced by the lithiasic treatment in a dose-dependent manner (Figure 1). This finding is similar to that obtained with a standardized extract of fenugreek seed [30] that was linked to an improvement in diuresis which resulted in the dissolution of the formed calculus and an interruption in the process of aggregation and deposition of the additional crystals. E-RT and the higher dose of EA-RT prevented the increase in serum urea, creatinine, and uric acid levels, probably by preserving a normal GFR. Electrolyte (calcium, phosphorus, K^+^, Na^+^, and Cl^−^) concentration enhancements were also inhibited by the RT extracts. Maintaining electrolytic balance may, therefore, result in the preservation of cell metabolism.

E-RT (2 g/kg) was the most effective in decreasing levels of urinary proteins and creatinine and restored EG/AC-induced low diuresis by improving the GFR. RT extracts also significantly reduced the levels of phosphorus, sodium, potassium, calcium, and uric acid. EA-RT (2 g/kg) had a lower beneficial effect and its action was limited to recovery of the GFR and the inhibition of stone formation. Of note, the lower dose (1 g/kg) of both EA-RT and E-RT was less effective than the higher dose 2 g/kg in inhibiting EG/AC-induced lithiasic effects as documented by biochemical changes in Table 1 and Table 2.

Histopathological analyses showed concordant results with biochemical changes. RT extracts’ dose consistently prevented the degenerative changes in kidney tissues that could be induced by EG/AC. Interestingly, RT extracts’ preventive effects depended on the type of extract. E-RT (2 g/kg) was the most effective in protecting from EG/AC-induced disorganization in kidney architecture even though no differences were found between EA-RT and E-RT at 2 g/kg regarding their protective effect on calcium oxalate urolithiasis formation. Results obtained with RT extracts were similar to those previously obtained with *Acorus calamus* ethanolic extract [22], *Peucedanum grande* hydroalcoholic extract [31], and cystone [32]. Biochemical and histopathological improvements in lithiasic animals observed after treatment with several plant extracts were proposed to be directly related to their antioxidant capacity. Antioxidants could have an important action in preventing the formation of the intrapapillary calcifications induced by oxidative stress that lead to papillary calculi formation [11].

To verify the antioxidant preventing effect hypothesis, the antioxidant properties of RT extracts were evaluated both in vitro and in vivo.

The in vitro assays showed a promising antioxidant activity of both E-RT and EA-RT extracts. As documented in the DPPH scavenging, reducing power and ß-carotene assays, the E-RT extract has a better antioxidant activity compared to EA-RT extract. However, this effect is less important than reference antioxidant agents (Table 5).

In vivo, EG treatment increased the level of MDA and significantly decreased the activity of the catalase enzyme in kidneys compared to the untreated control group (G1) (Figure 3). Indeed, elevated free radical production, as observed by an increase in MDA level due to EG ingestion in the formation of nephrolithiasis, confirms that kidney tissue is under oxidative stress. This hypothesis is strengthened by the report that patients with kidney stones have less activity of antioxidant enzymes with increased lipid peroxidation [33].

Oxidative damages, as reflected by higher lipid peroxidation (MDA) and lower antioxidant enzymes activity such as catalase, deteriorate kidney structure and functions as observed in calculi-induced rats. Antioxidant and reactive oxygen scavengers have been shown to be effective for protecting the kidney in animals [33]. Here, treatment with E-RT and EA-RT showed an increase in catalase enzyme activity and a decrease in MDA levels in a dose-dependent manner in kidney homogenates.

Overall, our results present evidence that *Rubia tinctorum* L. extracts exhibited a marked protective effect against oxidative stress both in vitro and in vivo.

*Rubia tinctorum* L. extract contain large amounts of antioxidants which can play an important role in adsorbing and neutralizing free radicals, quenching oxygen, or decomposing peroxides. This antioxidant activity may be due, in a large part, to the specific polyphenolic composition identified by the HPLC analysis (Table 6). Indeed, a positive correlation exists between total phenolic contents and the antioxidant capacity [34]. This correlation is confirmed here, as qualitative and quantitative analysis of the RT extracts’ composition showed that E-RT, which contained more various polyphenols at higher concentrations, has the most powerful antioxidant effect. The antioxidant potential of E-RT could be considered quite significant compared to the powerful phenol, gallic acid (this study) but it is important compared to extracts from other plants [35]. Analysis of the specific composition of each RT extract lead to the conclusion that E-RT antioxidant activity may be due to vanillin, rosmarinic acid, quercetin, catechin, syringic acid and cinnamic acid; while EA-RT antioxidant activity may be due to quercetin, cinnamic acid, vanillin and rutin. Several polyphenols with an antioxidant capacity were found to possess an antiurolithiasic preventing effect. As a matter of fact, quercetin antioxidant capacity was linked to a protective effect against oxidative stress associated with renal failure in the EG/AC model [29,36], to decreased oxidation of DNA bases [37] and to lead-induced DNA damage prevention and apoptosis [36]. There is also evidence that catechin has a preventive effect on renal calcium crystallization in vitro and in the EG model [38]. Vanillin and cinnamic acid may also contribute to an additional preventive effect as both compounds have antioxidant potential [39]. Therefore, it could be suggested that the superiority of E-RT over EA-RT related to their antioxidant action and prevention from urolithiasis is due to their richer polyphenolic constitution and their synergism.

These results are consistent with other studies that revealed that another species of the *Rubiaceae* family, *Rubia cordifolia* L., showed a great protective potential against different kidney and urinary disorders [21,40,41]. However, to our knowledge, for *Rubia tinctorum* L. this is the first time that a study reports such a preventive effective in urolithiasis.

## 4. Materials and Methods

### 4.1. Ethylene Glycol and Ammonium Chloride (EG/AC)-Induced Urolithiasis Model

#### 4.1.1. Animals

Thirty-six male Sprague Dawley rats (5–6 months) weighing between 180 and 260 g were procured from the animal facility of the Faculty of Sciences Semlalia, Marrakech, Morocco. All animals were initially acclimated in their cages for 3 days before the experiments (Figure 4). Experiments were conducted in accordance with internationally accepted standard guidelines for the use of laboratory animals described in the Scientific Procedures on Living Animals ACT 1986 (European Council directive: 86/609/EEC) and approved by Semlalia Faculty of Sciences Ethic Committee (protocol code SEML/PR/2019-PR12). The rats had free access to drinking water and daily chow and were kept under a controlled 12 h light/dark cycle at 25 ± 2 °C. All the experiments were performed in the morning according to Zimmermann et al. [42].

#### 4.1.2. Chemicals and Reagents

All chemicals used in this study were of analytical grade. Ethylene glycol (EG lot: BCBK 2604V), ethyl acetate (lot: SZBB184SV), gallic acid (lot: SLBM8746V), 1, 1-diphenyl-picrylhydrazyl (DPPH) (lot: 590790-379), and catechin (lot: WXBB6763V) were obtained from Sigma–Aldrich, Darmstadt, Germany. Ammonium chloride was purchased from Merck, Darmstadt, Germany (pro analysis, Lot: 9642642, ART. 1145). Ethanol (lot: 09L310512) and formaldehyde (lot: 11F200513) were procured from Prolabo VWR, Fontenay sous bois, France. PBS (Gibco, Ref 2285250) was obtained from Thermo Fischer Scientific, Waltham, MA USA 02451. Quercetin (≥95%) (Lot: SLBM7736V), rutin (≥95%) (Lot: VHS6475000V), ferulic acid (98%) (Lot: 1570363), rosmarinic acid (≥98%), cinnamic acid (≥99%) (Lot: SZBF048AV) were purchased from Sigma–Aldrich, (Germany). Vanillin (99.8%) (Lot: 080417CE), catechin (lot: WXBB6763V) and Syringic acid (≥95%) (Lot: 0478503CE) from Solvachim. Paraffin (lot: 1805052) was purchased from Leica biosystems, Nanterre, France and eosin (lot: 17060643) was purchased from Dako, Santa Clara, CA, USA. Kits used in this study for serum and urine dosage of urea (lot: 354868) (kinetic test with urease), creatinine (lot: 340775) (Jaffe reaction), uric acid (lot: 315894), phosphorus (lot: 325861), protein (lot: 307921) (Biuret Method), and Ca^2+^, Na^+^, K^+^, and Cl^−^ reagent set (lot: 04522320) (indirect potentiometric) were procured from ROCHE diagnostics, Indianapolis, IN 46256, USA.

#### 4.1.3. EG/AC Model Induction

The experimental urolithiasis rat model was induced according to the method described by Fan et al. [18]. Hyper-oxaluria and CaOx deposition in the kidney were induced by ethylene glycol (EG) being added to the drinking water to a final concentration of 0.75% for 10 days. To accelerate the process of lithiasis, 2% ammonium chloride (NH_4_CL or AC) was also added to the EG. Animals were weighed daily and divided into six groups of six each (Figure 4).

### 4.2. Preparation of RT Extracts and Assessment of Their Antilithiasic Effect

#### 4.2.1. Collection of Plant Material

*Rubia tinctorum* L. (RT), locally named “el foua”, was collected in June 2015 from the province of Azilal, Ait M’hamed village, geographic coordinates (31°51′00″ N, 6°30′48″ W), Morocco. The plant material was botanically classified and its correct botanical identification authenticated by Professor Ouhammou Ahmed Mohamed (Laboratory of Environment and Ecology (L 2 E, CNRST Associated Research Unit, URAC 32), Regional Herbarium MARK, Faculty of Sciences Semlalia, Cadi Ayad University, Marrakech, Morocco). A voucher specimen of the plant was deposited in the herbarium of the Semlalia Science Faculty, Cadi Ayad University (voucher number: 9825).

#### 4.2.2. Preparation of the Extracts

The dried RT roots were coarsely powdered. Then, 303 g of powder were packed into a soxhlet column and extracted with 70% *v*/*v* ethanol in water at 75–79 °C for 15 h. The extract obtained was evaporated at 45 °C. The ethanol extract was successively separated by a series of increasing polar solvents (hexan, ethyle acetate, butanol and distilled water) according to the method previously published [26,43]. The fraction thus obtained was concentrated with a rotary evaporator to obtain the following proportions of yield: 12.75% of RT-ethanolic extract and 5.28% of RT-ethyl acetate extract.

#### 4.2.3. Administration of the Extracts

A previous study did not show any toxicological effects of doses ranging between 0.5 to 5 g/kg; therefore, we selected the doses 1 and 2 g/kg [26].

After 3 days of acclimation, the 36 male Sprague Dawley rats were randomized in 6 groups of 6 individuals as follows (Figure 4):G1, the untreated control group (or vehicle control) received only distilled water during the ten days.G2, the lithiasic control group received distilled water supplemented with 0.75% ethylene glycol (EG) and 2% ammonium chloride (AC), during the ten days.G3 and G4 received distilled water supplemented with 0.75% ethylene glycol (EG) and 2% ammonium chloride (AC) and concomitantly, 1 mL/day of 1 or 2 g/kg of ethanolic extract of RT (E-RT), dissolved in distilled water, respectively, during the ten days.G5 and G6 received distilled water supplemented with 0.75% ethylene glycol (EG) and 2% ammonium chloride (AC) and concomitantly, 1 mL/day of 1 or 2 g/kg of ethyl acetate extract of RT (EA-RT), dissolved in distilled water, respectively, during the ten days.

### 4.3. Pathophysiological Parameters Evaluation

#### 4.3.1. Urine Collection and Analysis

On the 9th day, rats from each group were individually placed in metabolic cages for 24 h for urine collection. The calcium oxalate crystals’ shape and size were analyzed under light microscopy. After collection, urine samples were immediately analyzed for their volume, urea, creatinine, calcium, phosphorus, uric acid, Na, K, Cl, and total protein contents.

#### 4.3.2. Serum Analysis

After the 10-day experimental period, the rats were anesthetized with an i.p. chloral hydrate (10 mg/kg) injection. A blood volume of 1–2 mL was collected from the jugular vein in two tubes without any additive. Serum was separated by centrifugation (350 G, for 10 min) and used for biochemical (creatinine, urea, phosphorus, uric acid, calcium, Na, K, and Cl) dosage. Substrates, minerals, and electrolyte concentrations were determined enzymatically by standard methods with a biochemical automat (Cobas C311 analyzer, Roche Diagnostics GmbH, D-68298, Mannheim, Germany).

#### 4.3.3. Histopathological Analysis

After blood collection, the rats were sacrificed and both kidneys were carefully excised. Small slices of this freshly harvested tissue were fixed in a 10% formaldehyde solution buffered in 33 µM of sodium phosphate monobasic, dehydrated by serial ethanol solutions, diaphanized with ethanol-benzene, and enclosed in paraffin. Sections of 4 µm were sliced by a microtome (Leica, Microsystems Nussloch 6mbH, RM2125) before they were stained with haematoxylin and eosin and examined under a light microscope.

### 4.4. Antioxidant Activity

#### 4.4.1. DPPH Assay

DPPH assay was realized according to the method previously described [44]. An aliquot of 1 mL of various sample concentrations of E-RT and EA-RT were added to a volume of 2 mL of DPPH. The reaction mixture was well stirred and incubated for 20 min at room temperature in the dark. The absorbance of the extracts was measured at 512 nm using ethanol and ethyl acetate as control. Gallic acid was used as a 50% positive control. The percentage inhibition of the DPPH radical by the sample was calculated using the following equation:I% = [(A control − A sample)/A control] × 100(1)

“A control” is the absorbance of the control and “A sample” is the absorbance of the test sample. Extract concentration providing inhibition (IC_50_) was extracted from the absorbance concentration graph by plotting the inhibition percentage against extract concentration. All tests were carried out in triplicate.

#### 4.4.2. Lipid Peroxidation Determination

##### Tissue Samples Preparation

Kidneys tissues were homogenized with a polytron homogenizer in 1 mL of buffer containing PBS (Gibco pH = 7.4, Ref 2285250) and 0.04% Tween 80. Tissue samples were centrifuged at 10,000× *g* for 10 min. After centrifugation, supernatants were kept with sera at −20 °C until use.

##### Malondialdehyde Assay

The malondialdehyde (MDA) concentrations, as a product of lipid peroxidation, were evaluated by the thiobarbituric acid reactive substances method [45]. Briefly, the supernatant of the small intestine was mixed with 600 μL of 1% ortho-phosphoric acid and 200 μL of 0.6% thiobarbituric acid. They were then heated in a boiling water bath for 45 min. The completed reaction gave rise to a colorimetric product that will be extracted by the organic solvent n-butanol after centrifugation (2000× *g* during 15 min). The absorbance was read at 532 nm. MDA concentrations were expressed in terms of µmol of MDA per g of tissue (µmol/g of tissue) using a molar extinction coefficient of 1.56 × 10^5^ M^−1^ cm^−1^ for MDA.

#### 4.4.3. Catalase Activity Determination

The catalase activity (CAT) was initiated by adding 100 μL of each sample with 2.9 mL of the substrate hydrogen peroxide solution (H_2_O_2_, 0.2%) (Sigma–Aldrich, St. Louis, MO, USA) in a phosphate buffer (50 mM, pH 7.0) according to the method previously described [46]. Kinetics of degradation of H_2_O_2_ was followed for 3 min at 240 nm. The results were expressed in international unit per g of tissue (U/g of tissue).

#### 4.4.4. Reducing Power Assay

Reducing power was assessed according to the method previously described [47]. 1 mL of various sample concentrations was added to 2.5 mL of phosphate buffer (200 mM, pH 6.6) and 2.5 mL of potassium ferricyanide (1%). The resulting solution was incubated for 20 min at 50 °C, and after the incubation period 2.5 mL of 10% trichloroacetic acid (TCA) was mixed with the solution and then centrifuged at 200 G for 10 min. 2.5 mL of the upper layer solution was added to 2.5 mL of distilled water and 0.5 mL of 0.1% ferric chloride (FeCl_3_). The measurement of coloration formed by the reduction of Fe^3+^ at 700 nm was used to determine the sample concentration providing 0.5 of absorbance (IC_50_). BHT and quercetin were used as positive controls.

#### 4.4.5. β-Carotene Linoleic Acid Method

This is one of the rapid methods to screen antioxidants, which is mainly based on the principle that linoleic acid, which is an unsaturated fatty acid, gets oxidized by ‘‘reactive oxygen species’’ (ROS) produced by oxygenated water [48]. The products formed will initiate the β-carotene oxidation, which will lead to discoloration. Antioxidants decrease the extent of discoloration, which was measured at 434 nm, and the activity was measured.

β-carotene (0.5 mg) in 1 mL of chloroform was added to 25 µL of linoleic acid and 200 mg of tween-80 emulsified mixture. Chloroform was evaporated at 40 °C, 100 mL of distilled water saturated with oxygen was slowly added to the residue and the solution was vigorously agitated to form a stable emulsion. Four mL of this mixture was added into the test tubes containing 200 μL of the sample prepared in methanol. As soon as the emulsified solution was added to the tubes, zero-time absorbance was measured at 470 nm. The tubes were incubated for 2 h at 50 °C. Antioxidant activity is calculated as percentage of inhibition (I%) relative to the control using the following equation:I% = ((1 − (As − As120)/Ac − Ac120))(2)
where “As” was initial absorbance, “As120” was the absorbance of the sample at 120 min, “Ac” was initial absorbance of negative control and “Ac120” was the absorbance of the negative control at 120 min.

#### 4.4.6. Determination of Total Phenolic Content

The total phenolic contents of the E-RT and EA-RT was determined using the Folin–Ciocalteu method with a slight modification [49]. Sample and standard readings were made using a spectrophotometer (VR-200 spectrophotometer) at 760 nm against the reagent blank. The test sample (20 µL) was mixed with 1.16 mL of distilled water and 100 µL of Folin–Ciocalteu’s phenol reagent. After 8 min, 300 µL of saturated sodium carbonate solution (Na_2_CO_3_) 20% *w*/*v* in distilled water was added to the mixture. The reaction was kept in the dark for 40 min and after centrifugation, the absorbance of blue color from different samples was measured at 760 nm. The phenolic content was calculated as gallic acid equivalent (1–0.062 mg/mL, *Y =* 0.981 *x +* 0.003, *R*^2^
*=* 0.9999). All determinations were carried out in triplicate.

#### 4.4.7. Determination of Phenolic Compounds

The EA-RT and E-RT were analyzed for their content of flavonoids and polyphenols by the HPLC method. Chromatography separations were performed on a reversed-phase (RP-18) column (Agilent Technologies 250 mm × 4.6 mm, 5.0 µm), protected by a pre-column (Agilent Technologies RP-18, 10 mm × 4.6 mm). Both columns were placed in a column oven set at 25 °C. The HPLC system consisted of a pump (SCL-10), an automatic injector (SIL-10AD) and a detector (SPD 10A UV-visible) set at a spectrum beginning at 200 nm and ending at 700 nm (Shimadzu, Japan). Data collection and analysis were performed using Shimadzu LC Solution chromatography data station software. Two solvents were used with a constant flow rate of 1 mL/min and injection volume of 10 µL. Solvent A consisted of 5% acetonitrile and 95% water, whereas solvent B consisted of a phosphate buffer dissolved in water (pH 2.6). HPLC analysis was performed with the standards, followed by the RT extracts, and then the samples’ parameters were compared to the standards (syringic acid, vanillin, rosmarinic acid, catechin, cinnamic acid, quercetin, retinoic acid and ferulic acid).

### 4.5. Statistical Analysis

Statistical analyses were performed using the computer software Sigma Plot 12.5 for Windows. All data were expressed as mean ± standard error (SEM); *p* values less than 0.05, 0.01 and 0.001 were considered to be significant. Comparisons between different groups were performed using one-way analysis of variance (ANOVA). Significant differences between the control and experimental groups were assessed by Tukey’s test.

## 5. Conclusions

In conclusion, the concomitant oral administration of RT extracts prevented the development of EG/AC-induced urolithiasis in the rat. E-RT was found to be more efficient than EA-RT. The superiority of the E-RT preventive effect over EA-RT was associated with its more powerful antioxidant effect, resulting from its specific and rich polyphenol composition. E-RT was constituted by vanillin, rosmarinic acid, quercetin, catechin, syringic acid and cinnamic acid, whereas that of EA-RT contained specifically quercetin, cinnamic acid, vanillin and rutin. Taken together, our results suggest that RT could be an important alternative to anti-urolithiasis drugs due to its efficiency and minor side effects.

## Figures and Tables

**Figure 1 molecules-26-01005-f001:**
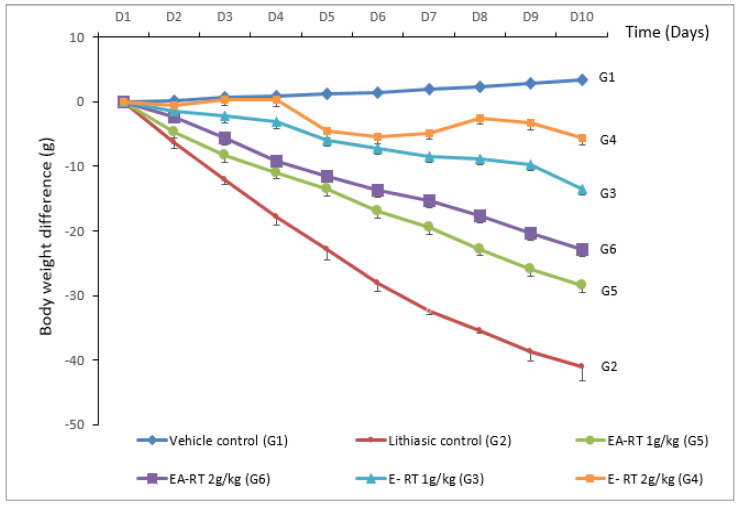
Cumulative body weight difference (mean ± SEM) in the 6 groups (6 rats/group) over the ten days of the treatment. Vehicle control (G1): received distilled water; Lithiasic control (G2): received 0.75% EG + 2% AC; E-RT 1 g/kg (G3): received concomitantly E-RT (1 g/kg) *plus* 0.75% EG + 2% AC; E-RT 2 g/kg (G4): received concomitantly E-RT (2 g/kg) *plus* 0.75% EG + 2% AC; EA-RT 1 g/kg (G5): received concomitantly EA-RT (1 g/kg) *plus* 0.75% EG + 2% AC; EA-RT 2 g/kg (G6): received concomitantly EA-RT (2 g/kg) *plus* 0.75% EG + 2% AC.

**Figure 2 molecules-26-01005-f002:**
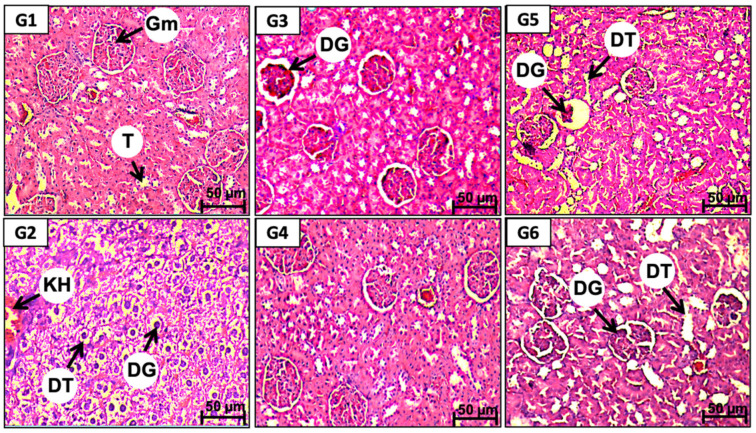
Representative microscopic images of hematoxylin-and-eosin-stained kidney sections observed under a light microscope from rats of: the vehicle control (**G1**), the lithiasic control (**G2**)**,** treated with E-RT at 1 g/kg (**G3**) and 2 g/kg (**G4**)**,** treated with EA-RT at 1 g/kg (**G5**) and 2 g/kg (**G6**). **Gm**: Glomeruli, **T**: Tubular, **DG**: Dilatation of glomeruli, **DT**: Dilatation of tubular, **KH**: kidney hemorrhage.

**Figure 3 molecules-26-01005-f003:**
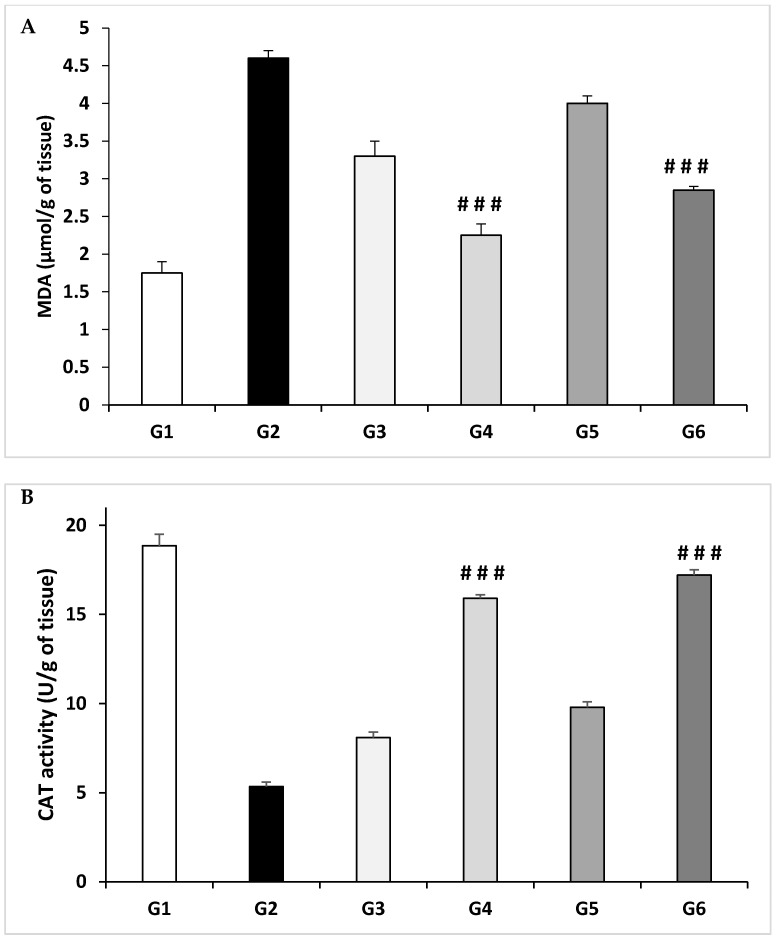
Evaluation of antioxidant activity in kidney homogenates using: (**A**) lipid peroxidation determination (MDA) and (**B**) catalase (CAT) enzyme activities. G1: Vehicle control; G2: Lithiasic control; G3: E-RT 1 g/kg; G4: E-RT 2 g/kg; G5: EA-RT 1 g/kg; G6: EA-RT 2 g/kg. The values shown are mean ± SEM of animals from each group (*n* = 6). ### *p* ≤ 0.001 vs. lithiasic control.

**Figure 4 molecules-26-01005-f004:**
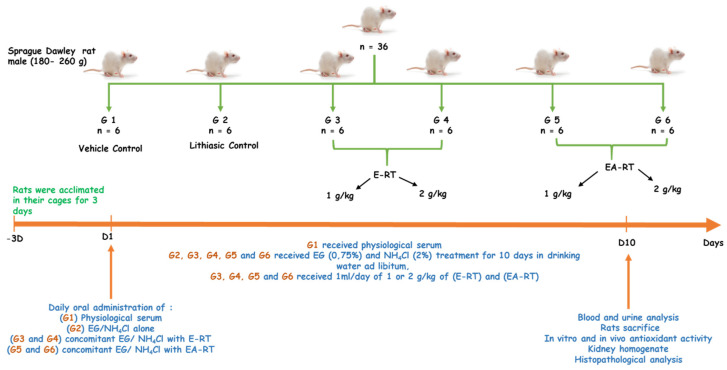
General experimental design diagram.

**Table 1 molecules-26-01005-t001:** Effects of ethanolic and ethyl acetate extracts of *Rubia tinctorum* L. (E-RT and EA-RT, respectively, 1 g/kg and 2 g/kg) on serum parameters on 10 days ethylene glycol and ammonium chloride (EG/AC)-induced urolithiasis rat model.

Parameters	G1 Vehicle Control	G2 Lithiasic Control	G3 E-RT 1 g/kg	G4 E-RT 2 g/kg	G5 EA-RT 1 g/kg	G6 EA-RT 2 g/kg
Urea (g/L)	0.26 ± 0.02	5.91 ± 0.17 ^b^	0.40 ± 0.01 ^a,c^	0.31 ± 0.02 ^a^	0.45 ± 0.07 ^a,b^	0.38 ± 0.04 ^a,d^
Creatinine (mg/L)	5.49 ± 0.21	39.51 ± 4.86 ^b^	5.65 ± 0.71 ^a^	5.16 ± 0.79 ^a^	7.08 ± 0.37 ^a^	6.93 ± 0.83 ^a^
Ca^2+^ (mg/L)	103.77 ± 1.01	119 ± 1.59 ^b^	106.33 ± 0.98 ^a^	104.50 ± 1.03 ^a^	109.5 ± 2.09 ^a^	107.7 ± 1.63 ^a^
Uric acid (mg/L)	10.71 ± 0.29	44.86 ± 3.00 ^b^	13.66 ± 0.95 ^a^	11.00 ± 1.03 ^a^	18.66 ± 1.28 ^a,b^	14.66 ± 1.60 ^a^
Phosphorus (mg/L)	83.34 ± 0.33	107.16 ± 1.74 ^b^	85.44 ± 1.39 ^a^	84.03 ± 3.40 ^a^	92.76 ± 3.23 ^a,d^	91.38 ± 1.26 ^a^
Na^+^ (mmol/L)	137.53 ± 0.29	182.55 ± 2.32 ^b^	139.20 ± 0.54 ^a^	137.63 ± 0.53 ^a^	141.68 ± 0.67 ^a,b^	139.97 ± 2.07 ^a^
Cl^−^ (mmol/L)	93.68 ± 0.72	151.55 ± 1.30 ^b^	94.49 ± 1.17 ^a^	93.00 ± 0.81 ^a^	96.48 ± 0.76 ^a^	95.34 ± 0.96 ^a^
K^+^ (mmol/L)	8.24 ± 0.07	12.27 ± 0.34 ^b^	8.55 ± 0.07 ^a^	8.36 ± 0.09 ^a^	9.11 ± 0.39 ^a^	8.56 ± 0.25 ^a^

Values are mean ± standard error (SEM), *n* = 6, ^a^
*p <* 0.001 significantly different compared to the lithiasic control, ^b^
*p <* 0.001, ^c^
*p* < 0.01, and ^d^
*p* < 0.05 significantly different compared to the vehicle control. E-RT: ethanolic extract of RT; EA-RT: ethyl acetate extract of RT.

**Table 2 molecules-26-01005-t002:** Effects of ethanolic and ethyl acetate extracts of *Rubia tinctorum* L. (E-RT and EA-RT, respectively), at different concentrations (i.e., 1 g/kg and 2 g/kg) on urine parameters on 10 days ethylene glycol and ammonium chloride (EG/AC)-induced urolithiasis rat model.

Parameters	G1 Vehicle Control	G2 Lithiasic Control	G3 E-RT 1 g/kg	G4 E-RT 2 g/kg	G5 EA-RT 1 g/kg	G6 EA-RT 2 g/kg
Urea (g/L)	33.96 ± 0.95	317.90 ± 0.95 ^b^	27.15 ± 1.85 ^a,c^	34.12 ± 1.27 ^a^	28.53 ± 1.64 ^a^	25.96 ± 0.26 ^a,c^
Creatinine (mg/L)	27.75 ± 1.66	116.28 ± 4.02 ^b^	35.98 ± 1.53 ^a,c^	31.32 ± 1.07 ^a^	38 ± 2.31 ^a,b^	35.21 ± 1.66 ^a,d^
Ca^2+^ (mg/L)	74.36 ± 1.36	217.08 ± 3.73 ^b^	98.16 ± 2.65 ^a,b^	73.57 ± 1.76 ^a^	83.95 ± 3.79 ^a^	77.66 ± 1.56 ^a^
Uric acid (mg/L)	58.33 ± 3.48	230.5 ± 10.79 ^b^	101.83 ± 0.70 ^a,b^	70 ± 3.14 ^a^	121.83 ± 0.94 ^a,b^	149.83 ± 0.98 ^a,b^
Phosphorus (mg/L)	12.83 ± 0.60	299.41 ± 1.47 ^b^	34.98 ± 0.78 ^a,b^	17.16 ± 0.47 ^a,d^	234.25 ± 0.93 ^a,b^	163.16 ± 0.94 ^a,b^
Na^+^ (mmol/L)	8.33 ± 0.21	18.33 ± 0.49 ^b^	10 ± 0.57 ^a^	9.83 ± 1.16 ^a^	8.66 ± 0.49 ^a^	12 ± 0.57 ^a,c^
K^+^ (mmol/L)	6.13 ± 0.16	196.03 ± 0.81 ^b^	42.74 ± 0.73 ^a,b^	12.67 ± 0 ^a,b^	22.56 ± 0.29 ^a,b^	17.38 ± 0.16 ^a,b^
Cl^−^ (mmol/L)	36.45 ± 0.59	132.31 ± 0.75 ^b^	99.42 ± 0.50 ^a,b^	56.66 ± 0.34 ^a,b^	107.55 ± 2.61 ^a,b^	68.94 ± 0.98 ^a,b^
Protein (mg/L)	0.28 ± 0.01	1.03 ± 0.04 ^b^	0.36 ± 0.06 ^a^	0.11 ± 0.01 ^a,d^	0.78 ± 0.02 ^a,b^	0.76 ± 0.01 ^a,b^

Values are mean ± SEM, *n* = 6; ^a^
*p* < 0.001 significantly different compared to the lithiasic control. ^b^
*p* < 0.001, ^c^
*p* < 0.01, and ^d^
*p* < 0.05 significantly different compared to the vehicle control.

**Table 3 molecules-26-01005-t003:** Histopathological changes in the kidneys of urolithiasic rats treated or not treated with RT extracts.

	G1 Vehicle Control	G2 Lithiasic Control	G3 E-RT 1 g/kg	G4 E-RT 2 g/kg	G5 EA-RT 1 g/kg	G6 EA-RT 2 g/kg
Cloudy swelling	-	+++	-	-	++	+
Infiltration of mononuclear cells	-	+++	-	-	++	+
Kidney hemorrhage	-	+++	+	-	++	+
Morphological disorganization of tubules and glomeruli	-	+++	+	+	++	++

The average results of evaluation in each group (*n* = 6) were scored: -: no modifications (no cellular damage); +: slight alteration; ++: moderate alteration; +++: marked alteration.

**Table 4 molecules-26-01005-t004:** IC_50_ (µg/mL) values of E-RT and EA-RT extracts compared to gallic acid using the DPPH assay (*n* = 3).

	IC_50_ (µg/mL)
Gallic acid (GA)	64.50 ± 0.70
E-RT	156.44 ± 35.76 ^a^
EA-RT	206.23 ± 90.68 ^a^

Values are mean ± standard error (SEM), ^a^
*p* < 0.001 significantly different compared to gallic acid.

**Table 5 molecules-26-01005-t005:** IC_50_ (µg/mL) values of E-RT and EA-RT extracts compared to reference antioxidants using the reducing power and ß-carotene assays (BHT and quercetin) (*n* = 3).

	IC_50_ (µg/mL) Reducing Power	IC_50_ (µg/mL) β-Carotene
Butylated hydroxytoluene (BHT)	0.12 ± 0.01	0.74 ± 0.02
Quercetin	0.07 ± 0.01	4.39 ± 0.05
E-RT	2.44 ± 0.02 ^a,b^	75.61 ± 3.33 ^a,b^
EA-RT	4.66 ± 0.04 ^a,b^	101.64 ± 5.41 ^a,b^

Values are expressed as mean ± standard error (SEM), ^a^
*p* < 0.001 significantly different compared to BHT, ^b^
*p* < 0.001, significantly different compared to quercetin.

**Table 6 molecules-26-01005-t006:** Concentrations of the main phenolic compounds identified in E-RT and EA-RT expressed in mg GAE/100 g DM.

	Phenolic Compounds	Concentration (mg GAE/100 g DM)	Retention Time
E-RT	Syringic acid	13.79 ± 0.12	17.0
	Vanillin	26.74 ± 0.15	28.8
	Rosmarinic acid	15.61 ± 0.20	29.4
	Cinnamic acid	14.19 ± 0.10	41.8
	Catechin	13.78 ± 0.09	44.1
	Quercetin	14.90 ± 0.21	45.1
EA-RT	Rutin	12.32 ± 0.08	21.5
	Ferulic acid	12.14 ± 0.13	23.3
	Vanillin	12.32 ± 0.11	28.8
	Cinnamic acid	12.43 ± 0.09	41.8
	Quercetin	15.81 ± 0.12	45.1

Values are expressed as mean ± standard error (SEM) (*n* = 3).

## Data Availability

The data presented in this study are available upon reasonable request from the authors.

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
