# Peer review of "Antioxidant and Polyphenol-Rich Ethanolic Extract of *Rubia tinctorum* L. Prevents Urolithiasis in an Ethylene Glycol Experimental Model in Rats"

_molecules, 2021, doi:10.3390/molecules26041005_

Round 1

Reviewer 1 Report

The paper entitled: "Antioxidant and Polyphenol-Rich Ethanolic Extract of Rubia Tinctorum L. Prevents Urolithiasis in an Ethylene Glycol Experimental Model in Rats" by Fatima Zahra Marhoume et al. is a quite well organized manuscript.

The authors presented an interresting view of the role of Rubia tinctorum L and it can be accepted for publication after major revision.

Author Response

Authors’ response

We would like to thank Reviewer 1 for his valuable comments, time and effort in reading our manuscript.

The manuscript has been deeply revised and we believe it has been greatly improved. The revisions are highlighted in blue in the manuscript.

We hope that the changes made will suit Reviewer 1.

Reviewer 2 Report

It was my pleasure to read the manuscript. The description of lab work and results is excellent. The conclusions supported. Figures and tables well balanced.

There are, however, two minor comments:

  1. Figure 2 (p5) is not very clear or aligned. The space between the six boxes is uneven. An additional problem, that can also be easily fixed is the problem of contrast. The crystals are very hard to see in the image, so adding black arrows or circling them, or enhancing the contrast between backdrop and crystals will significantly improve their visibility and the usability of figure 2
  2. In M&M, there are some issues with the extract preparation method. In this manuscript description, the isolation of extract was done with only one solvent (70% ethanol) and later it was fractioned in some way into ethanol and ethyl acetate fractions. Authors reference more details in the paper under reference no.42, but in that paper, the authors used three solvents with different polarity consecutively on the plant material, and crude fractions were obtained in this way. Please explain in more detail how was the extraction and separation done in this case. 

Author Response

It was my pleasure to read the manuscript. The description of lab work and results is excellent. The conclusions supported. Figures and tables well balanced.

There are, however, two minor comments:

Figure 2 (p5) is not very clear or aligned. The space between the six boxes is uneven. An additional problem, that can also be easily fixed is the problem of contrast. The crystals are very hard to see in the image, so adding black arrows or circling them, or enhancing the contrast between backdrop and crystals will significantly improve their visibility and the usability of figure 2

Authors’ response

As Figure2 is not very clear and because of comment from another reviewers, who felt it was not informative, we decided to remove it from the manuscript and add it to supplementary data.

In M&M, there are some issues with the extract preparation method. In this manuscript description, the isolation of extract was done with only one solvent (70% ethanol) and later it was fractioned in some way into ethanol and ethyl acetate fractions. Authors reference more details in the paper under reference no.42, but in that paper, the authors used three solvents with different polarity consecutively on the plant material, and crude fractions were obtained in this way. Please explain in more detail how was the extraction and separation done in this case. 

Authors’ response

Reference 42 has been amended by Elhabazi, Aboufatima et al. 2006, and Marhoume, Laaradia et al. 2019. The extraction method was adapted from reference 42. Therefore, we prefer replace reference 42 with two references from our laboratory that better describe the extraction method.

Authors thank reviewer for its thoughtful remarks, time and efforts spent in reading and revising our manuscript. We made substantial changes that appear highlighted in manuscript.

Reviewer 3 Report

A work entitled “Antioxidant and Polyphenol-Rich Ethanolic Extract of Rubia tinctorum L. Prevents Urolithiasis in an Ethylene Glycol Experiimental Model in Rats” contains novel data, and therefore, it could be interesting for readers. However, the present form of the manuscript needs a revision and several improvements. Main concerns are presented below. Because the text contains numerous errors or ambiguities, most of my comments was included directly into the manuscript text. Please, find the attached file with the remaining suggestions concerning the quality of data presentation and the required corrections.

Could the authors, please, explain why the antioxidant tests were performed using different reference compounds? The same reference antioxidants both in the DPPH, reducing power and beta-carotene assays seem to be more logical.

Please, correct the sentences where quercetin is described as synthetic antioxidant (lines 199-200).

The description of OY axis in the figure 4 should be corrected. Please, verify the used units.

Please, verify the calculation procedure and the equation formula in the DPPH Assay section.

Author Response

A work entitled “Antioxidant and Polyphenol-Rich Ethanolic Extract of Rubia tinctorum L. Prevents Urolithiasis in an Ethylene Glycol Experiimental Model in Rats” contains novel data, and therefore, it could be interesting for readers. However, the present form of the manuscript needs a revision and several improvements. Main concerns are presented below. Because the text contains numerous errors or ambiguities, most of my comments was included directly into the manuscript text. Please, find the attached file with the remaining suggestions concerning the quality of data presentation and the required corrections.

Authors’ response

Authors are grateful to reviewer for its deep review of the manuscript. The comments were fair and relevant. We feel that the manuscript is highly improved.

Could the authors, please, explain why the antioxidant tests were performed using different reference compounds? The same reference antioxidants both in the DPPH, reducing power and beta-carotene assays seem to be more logical.

Authors’ response

DPPH allows testing of both lipophilic and hydrophilic compounds while other methods are restricted in the nature of antioxidants that they can quantify. The β-carotene-linoleate bleaching (BCB) method employs an emulsified lipid and therefore used to investigate lipophilic antioxidants. If polar compounds such as rosmarinic acid, etc., are tested by the BCB method, they would be considered as weak antioxidants, therefore the reference compounds that we used were was BHT and quercetin (Koleva, van Beek et al. 2002, Kulisic, Radonic et al. 2004).

Please, correct the sentences where quercetin is described as synthetic antioxidant (lines 199-200).

Authors’ response

Sentence was amended in the manuscript. Now sentence is:

“it is less important than reference antioxidant agents such BHT and quercetin”

The description of OY axis in the figure 4 should be corrected. Please, verify the used units.

Authors’ response

The units were verified. For MDA the correct unit is µmol/g.

Please, verify the calculation procedure and the equation formula in the DPPH Assay section.

Authors’ response

The equation formula was verified: I% = [(A control – A sample) / A control]*100

Polyphenol content

Authors apologize for the mistake, after verifying HPLC chromatograms and standards it’s not “retinoic acid” (not a polyphenol) it’s rutin which is a polyphenol.

Reviewer 4 Report

Comments to the manuscript molecules-1091622 "Antioxidant and Polyphenol-Rich Ethanolic Extract of Rubia Tinctorum L. Prevents Urolithiasis in an Ethylene Glycol Experimental Model in Rats".

Authors propose the report of an experiment aimed to demonstrate the effectiveness of ethanol or ethyl-acetate extracts of the Rubia tinctorum root tissues to prevent urolithiasis induced in rats. The antioxidant properties of the extracts were also measured by some in vitro and in vivo tests, and the composition of some phenolic compounds was determined by HPLC. The state of the art and research objectives were clearly presented in the introduction. The experiment was correctly designed and described in the materials and methods section. The data treatment was appropriated and the presentation of the results sufficiently clear, as well as the discussion. The research represents a good contribution to knowledge in the field of studies. In my opinion, the manuscript is suitable for publication after some minor changes in the presentation of the results.

Please check the following suggestions.

1) Figure 3: please use white characters and lines in order to improve the readability of the notes inside the pictures. The black is badly readable on the red ground.

2) Table 4 and Table 5: please add to the average data the result of a mean separation test (letters) in order to evaluate the significance of the differences.

3) Table 6: please add to the average data ±SEM or another variability indicator.

Author Response

Comments to the manuscript molecules-1091622 "Antioxidant and Polyphenol-Rich Ethanolic Extract of Rubia Tinctorum L. Prevents Urolithiasis in an Ethylene Glycol Experimental Model in Rats".

Authors propose the report of an experiment aimed to demonstrate the effectiveness of ethanol or ethyl-acetate extracts of the Rubia tinctorum root tissues to prevent urolithiasis induced in rats. The antioxidant properties of the extracts were also measured by some in vitro and in vivo tests, and the composition of some phenolic compounds was determined by HPLC. The state of the art and research objectives were clearly presented in the introduction. The experiment was correctly designed and described in the materials and methods section. The data treatment was appropriated and the presentation of the results sufficiently clear, as well as the discussion. The research represents a good contribution to knowledge in the field of studies. In my opinion, the manuscript is suitable for publication after some minor changes in the presentation of the results.

Please check the following suggestions.

Authors thank reviewer for its fair and well considered comments. Changes to improve the article appear highlighted in manuscript.

  • Figure 3: please use white characters and lines in order to improve the readability of the notes inside the pictures. The black is badly readable on the red ground.

Authors’ response

Reviewer is right. Some changes have been made in the figure to improve its readability. We hope that the changes will suit reviewer.

  • Table 4 and Table 5: please add to the average data the result of a mean separation test (letters) in order to evaluate the significance of the differences.

Authors’ response

Statistical tests have been made for results of Table 4 and 5.

  • Table 6: please add to the average data ±SEM or another variability indicator.

Authors’ response

Data ±SEM has been added from three separate HPLC analysis. Data have been amended.

Reviewer 5 Report

Rubia tinctorum, the common madder or dyer's madder, is a perennial plant species in the Rubiaceae. Roots of R. tinctorium have been used in traditional medicine in the Caucasus region and Middle Asia as a diuretic. Extracts of  Rubia have been reported to decrease the tension and increase the amplitude of renal pelvic and ureter contractions. This extract promotes the motility of stones and their elimination from the kidney and the urinary tract.

While reading the manuscript, I had some questions.

  1. Madder dye contains anthracene derivatives of the alizarin, namely ruberitric acid, which has an antispasmodic, diuretic, antimicrobial effect and is used to treat urolithiasis. Please justify the choice of polyphenols in gallic acid aquivalents as a parameter that determines the activity of madder dye extract for the treatment of urolithiasis.
  2. Madder dye is widely used in various traditional medicine, such as European, Russian (e.g. https://doi.org/10.1016/j.jep.2020.113685) and others. Please discuss the data on the widespread.
  3. The identification of each compounds (syringic acid, vanillin, rosmarinic acid, catechin, cinnamic acid, quercetin, retinoic acid and ferulic acid) must be confirmed by the addition of the reference compound. Quantification should be done by absolute calibration. Please provide linear equations for all identified compounds.
  4. Please explain why the total polyphenols by Folin-Chocaltey method (section 2.3.2) is many times less than the sum of the identified phenolic compounds (Table 6).
  5. There is no conclusion of the bioethical commission (section 4.1.1).
  6. Please indicate in section 4.1.2 the source of reference materials for the quantitative determination of phenolic compounds.
  7. Please provide a justification for choosing a dose of 1 and 2 g / kg extract for administration to animals. What is the solubility of the extracts if they were introduced as a solution? What was the volume of administration to the animals? Weren't the humane principles of working with animals violated? How were the extracts administered to animals?
  8. In section 4.4, please indicate the literature sources of all methods used.
  9. Figure 2 is not informative. Consider to delete.

Author Response

Rubia tinctorum, the common madder or dyer's madder, is a perennial plant species in the Rubiaceae. Roots of R. tinctorium have been used in traditional medicine in the Caucasus region and Middle Asia as a diuretic. Extracts of Rubia have been reported to decrease the tension and increase the amplitude of renal pelvic and ureter contractions. This extract promotes the motility of stones and their elimination from the kidney and the urinary tract.

While reading the manuscript, I had some questions.

Madder dye contains anthracene derivatives of the alizarin, namely ruberitric acid, which has an antispasmodic, diuretic, antimicrobial effect and is used to treat urolithiasis. Please justify the choice of polyphenols in gallic acid aquivalents as a parameter that determines the activity of madder dye extract for the treatment of urolithiasis.

Authors’ response

It is well established that Kidney stones, also known as calcium oxalate (CaOx) nephrolithiasis induce oxidative stress (Yuan, Zhang et al. 2020). As polyphenols have great antioxidant activity, then protective effect of R. tinctorium can be attributed to polyphenols.

Furthermore, Gallic acid is usually used as a reference chemical in total phenolic test. Gallic acid fullfil the criteria to consider in the choice of the phenolic acid to be used as standard i.e.,: the stability, the solubility in the solvent used and the abundance in the extracts of RT.

Madder dye is widely used in various traditional medicine, such as European, Russian (e.g. https://doi.org/10.1016/j.jep.2020.113685) and others. Please discuss the data on the widespread.

Authors’ response

Authors thank reviewer for this relevant reference that has been added to manuscript (Ref number 13 in manuscript). Although authors make efforts to cite all recent references, we missed this reference as this reference is very recent (March, 2021).

The identification of each compounds (syringic acid, vanillin, rosmarinic acid, catechin, cinnamic acid, quercetin, retinoic acid and ferulic acid) must be confirmed by the addition of the reference compound. Quantification should be done by absolute calibration. Please provide linear equations for all identified compounds.

Authors’ response

Reference of each compound have been added to the manuscript (Section 4.1.2. Chemicals and Reagents)

As far as concerns compounds quantification, there are different methods that can be used. In our laboratory we use the method previously described by publications from our team and by others (Sun, Jin et al. 2017, Oufquir, Ait Laaradia et al. 2020)

U.S. EPA. 2000. "Method 8318A (SW-846): N-Methylcarbamates by High Performance Liquid Chromatography (HPLC)," Revision 1. Washington, DC.

Please explain why the total polyphenols by Folin-Chocaltey method (section 2.3.2) is many times less than the sum of the identified phenolic compounds (Table 6).

Authors’ response

Whereas total phenol contents, by the Folin–Ciocalteu method, was determined in a volume of 20 µl, concentrations of the main phenolic compounds using HPLC were in mg GAE/100 g DM.

There is no conclusion of the bioethical commission (section 4.1.1).

Authors’ response

Rats were treated in compliance with the guidelines of the UCAM related to the ethical evaluation and authorization of projects using animals for experiments. All procedures were in accordance to the European decree NOR: AGRG1238767A, 2013. All the efforts were made to minimise the numbers of animals and suffering.

Please indicate in section 4.1.2 the source of reference materials for the quantitative determination of phenolic compounds.

Authors’ response

Section 4.1.2 is dedicated for chemicals and reagents. Reference materials is added in each section when used.

Please provide a justification for choosing a dose of 1 and 2 g / kg extract for administration to animals. What is the solubility of the extracts if they were introduced as a solution? What was the volume of administration to the animals? Weren't the humane principles of working with animals violated? How were the extracts administered to animals?

Authors’ response

The doses used were largely bellow potential toxic doses that were determined according to:

  • The OECD Guidelines for Acute Oral Toxicity OECD 423
  • A study previously published (Marhoume, Laaradia et al. 2019)

The extracts are soluble in water.

The volume of administration varies according to the weight of the rat following the formula:

10 ml ->1000 g

As mentioned above and in materials and methods section, great care was taken to minimize animals suffering and reduce the number of animals used. Rats were treated in compliance with the guidelines of the UCAM related to the ethical evaluation and authorization of projects using animals for experiments.

The extracts were administrated orally

In section 4.4, please indicate the literature sources of all methods used.

Authors’ response

References of all methods have been added to suitable sections.

Figure 2 is not informative. Consider to delete.

Authors’ response

Figure 2 has been removed from manuscript and added to supplementary data.

Authors thank reviewer for time and effort reading our manuscript and giving relevant comments and thoughtful remarks.

References

Elhabazi, K., R. Aboufatima, A. Benharref, A. Zyad, A. Chait and A. Dalal (2006). "Study on the antinociceptive effects of Thymus broussonetii Boiss extracts in mice and rats." J Ethnopharmacol 107(3): 406-411.

Koleva, II, T. A. van Beek, J. P. Linssen, A. de Groot and L. N. Evstatieva (2002). "Screening of plant extracts for antioxidant activity: a comparative study on three testing methods." Phytochem Anal 13(1): 8-17.

Kulisic, T., A. Radonic, V. Katalinic and M. Milos (2004). "Use of different methods for testing antioxidative activity of oregano essential oil." Food Chemistry 85(4): 633-640.

Marhoume, F. Z., M. A. Laaradia, Y. Zaid, J. Laadraoui, S. Oufquir, R. Aboufatima, A. Chait and A. Bagri (2019). "Anti-aggregant effect of butanolic extract of Rubia tinctorum L on platelets in vitro and ex vivo." J Ethnopharmacol 241: 111971.

Oufquir, S., M. Ait Laaradia, Z. El Gabbas, K. Bezza, J. Laadraoui, R. Aboufatima, Z. Sokar and A. Chait (2020). "Trigonella foenum-graecum L. Sprouted Seed Extract: Its Chemical HPLC Analysis, Abortive Effect, and Neurodevelopmental Toxicity on Mice." Evid Based Complement Alternat Med 2020: 1615794.

Sun, L., H. Y. Jin, R. T. Tian, M. J. Wang, L. N. Liu, L. P. Ye, T. T. Zuo and S. C. Ma (2017). "A simple method for HPLC retention time prediction: linear calibration using two reference substances." Chin Med 12: 16.

Yuan, H., J. Zhang, X. Yin, T. Liu, X. Yue, C. Li, Y. Wang, D. Li and Q. Wang (2020). "The protective role of corilagin on renal calcium oxalate crystal-induced oxidative stress, inflammatory response, and apoptosis via PPAR-γ and PI3K/Akt pathway in rats." Biotechnol Appl Biochem.

Round 2

Reviewer 1 Report

The paper entitled: "Antioxidant and Polyphenol-Rich Ethanolic Extract of Rubia Tinctorum L. Prevents Urolithiasis in an Ethylene Glycol Experimental Model in Rats" by Fatima Zahra Marhoume et al. is a quite well presented manuscript. It has been properly revised and it can be accepted for publication.

Reviewer 5 Report

The authors responded to my comments and made the necessary corrections. I have no more questions.